# Early Changes in Acute Myocardial Infarction in Pigs: Achieving Early Detection with Wearable Devices

**DOI:** 10.3390/diagnostics13061006

**Published:** 2023-03-07

**Authors:** Ke Li, Marco Morales-Garza, Cristiano Cardoso, Angel Moctezuma-Ramirez, Atandra Burman, Jitto Titus, Abdelmotagaly Elgalad, Emerson Perin

**Affiliations:** 1Center for Preclinical Research, The Texas Heart Institute, Houston, TX 77030, USA; 2Remote Cardiac Enablement (RCE Inc.), Carlsbad, CA 92008, USA; 3Center for Clinical Research, The Texas Heart Institute, Houston, TX 77030, USA

**Keywords:** wearable device, acute myocardial infarction, early detection, animal model

## Abstract

We examined the changes in variables that could be recorded on wearable devices during the early stages of acute myocardial infarction (AMI) in an animal model. Early diagnosis of AMI is important for prognosis; however, delayed diagnosis is common because of patient hesitation and lack of timely evaluations. Wearable devices are becoming increasingly sophisticated in the ability to track indicators. In this study, we retrospectively reviewed the changes in four variables during AMI in a pig model to assess their ability to help predict AMI onset. AMI was created in 33 pigs by 90-min balloon occlusion of the left anterior descending artery. Blood pressure, EKG, and lactate and cardiac troponin I levels were recorded during the occlusion period. Blood pressure declined significantly within 15 min after balloon inflation (mean arterial pressure, from 61 ± 8 to 50 ± 8 mmHg) and remained at this low level. Within 5 min of balloon inflation, the EKG showed ST-elevation in precordial leads V1–V3. Blood lactate levels increased gradually after occlusion and peaked at 60 min (from 1.48 to 2.53 mmol/L). The continuous transdermal troponin sensor demonstrated a gradual increase in troponin levels over time. Our data suggest that significant changes in key indicators (blood pressure, EKG leads V1–V3, and lactate and troponin levels) occurred at the onset of AMI. Monitoring of these variables could be used to develop an algorithm and alert patients early at the onset of AMI with the help of a wearable device.

## 1. Introduction

Acute myocardial infarction (AMI) is a major cause of death and disability worldwide. The risk of death is highest in the first hours after onset of chest pain [1]. Therefore, an early and accurate diagnosis of AMI in patients with chest pain is of paramount importance for initiating effective treatment and improving outcome [2]. Currently, the essential indicators for AMI diagnosis are clinical symptoms, electrocardiographic (ECG) changes, and the rise or fall in cardiac troponin levels. However, these indicators typically require a physician’s attention and the use of professional devices in the hospital for evaluation. Patients often misunderstand chest pain or hesitate to go to the hospital [3,4]. Hence, diagnosis and early treatment can be easily delayed [5]. Novel wearable devices have become increasingly popular and advanced and can be used to track key indicators and actively alert patients who are at high risk of AMI. This strategy may facilitate AMI diagnosis and potentially enable personalized treatment in the future.

A series of physiological changes occur when an AMI happens. Capturing these changes at the exact moment of symptom onset is difficult [6]. In the US, the time from symptom onset to hospital arrival in patients with an AMI ranges from 1.5 to 6 h [7]; the median time is about 3.5 h, with most patients presenting to the Emergency Department more than 3 h after symptom onset [3,8]. To better understand the changes that occur at the initiation of an AMI, we retrospectively reviewed the hemodynamic and biological changes in pigs in which an AMI was induced at our preclinical center.

## 2. Materials and Methods

We reviewed the data on pigs who underwent AMI induction via balloon occlusion across multiple protocols that had been approved by the institutional animal care and use committee at The Texas Heart Institute. In the AMI protocol, the pig was placed in the supine position under general anesthesia. A 5F hockey stick coronary-guiding catheter (Convey, Boston Scientific, Natick, MA, USA) was percutaneously advanced through the aorta and placed at the left main coronary ostium without full engagement under the guidance of C-Arm fluoroscopy. Then, a 20 mm long monorail angioplasty balloon (Emerge, Boston Scientific, Natick, MA, USA) of 2.25 to 3.5 mm diameter, depending on the diameter of the left anterior descending artery (LAD) (3 mm typically), was placed into the LAD between the first and second diagonal branch over a 0.014″ straightforward workhorse guidewire (Choice Floppy, Boston Scientific, Natick, MA, USA). The balloon was inflated at the minimum pressure needed to ensure occlusion of the LAD distal to the balloon for 90 min. Complete coronary occlusion was confirmed by the absence of distal flow of contrast.

For the current study, we reviewed the anesthesia records for all pigs, which provided the heart rate (HR) and blood pressure reading every 15 min. Of the 33 pigs in our study, 16 had 12-lead ECG recordings at different time points before and during balloon inflation. Eight pigs were evaluated for lactate (IDEXX Laboratories, Westbrook, ME, USA) at multiple time points. In these 8 pigs, a transdermal troponin sensor (Tropsensor; Remote Cardiac Enablement, Carlsbad, CA, USA) had also been attached to their ears for non-invasive troponin readings; traditional troponin (IDEXX Laboratories, Westbrook, ME, USA) and high-sensitivity troponin (enzyme-linked immunosorbent analysis [ELISA]) analyses were completed at preset time points. All continuous values were expressed as mean ± standard deviation for data analysis. To compare continuous variables at different time points, we used analysis of variance (ANOVA); a *p* value < 0.05 was considered significant.

## 3. Results

This section may be divided by subheadings. It should provide a concise and precise description of the experimental results, their interpretation, as well as the experimental conclusions that can be drawn.

### 3.1. Hemodynamic Data

Figure 1 shows the change in the hemodynamic variables of HR, systolic arterial pressure (SAP), diastolic arterial pressure (DAP), and mean artery pressure (MAP) before and after LAD occlusion. All variables except HR decreased significantly from baseline to 15 min after balloon occlusion (*p* < 0.05): SAP, from 89 ± 14 mmHg to 71 ± 12 mmHg; DAP, from 46 ± 6 mmHg to 40 ± 6 mmHg; and MAP, from 61 ± 8 mmHg to 50 ± 8 mmHg. The decreased levels were maintained until the end of the occlusion.

### 3.2. ECG

Figure 2 shows a representative 12-lead ECG of a pig 5 min after balloon occlusion of the LAD. ST elevation was seen in the precordial leads (V1–V3) as early as 5 min after inflation.

### 3.3. Biomarkers

#### 3.3.1. Blood Lactate

Blood lactate levels gradually increased after occlusion, reaching a peak level around 60 min (Figure 3; from 1.48 ± 0.19 mmol/L to 2.53 ± 1.45 mmol/L). Increased levels were maintained even at the 90 min time point (2 ± 0.85 mmol/L). However, the change over time was not statistically significant as indicated by ANOVA analysis.

#### 3.3.2. Troponin

No changes were detected in blood troponin levels during the occlusion by either traditional (iSTAT) or high-sensitivity troponin (ELISA) analyses. However, the Tropsensor showed a trend of a continuous increased optical signal from the point of inflation closely after the onset of ST elevation. The Tropsensor signal was flat before inflation, which serves as a reference (Figure 4).

## 4. Discussion

In this retrospective study, we reviewed the hemodynamic data from pigs with AMI at our institution. Specifically, our results provide valuable information on the early changes that occur in the first 90 min after the start of an AMI. We observed several important changes in that early time period. Blood pressure dropped immediately in the first 15 min after balloon occlusion, and the ST elevation on ECG was seen within 5 min after ischemia in the corresponding precordial leads. The upward trend in AMI biomarkers was more gradual over time after AMI. However, we did not see a change in HR; the absence of tachycardia in our study may relate to the general anesthesia and antiarrhythmic medication used during the procedure. These findings provide a clear picture of the early changes seen in AMI and are consistent with the pathophysiological changes seen after AMI in humans (Figure 5).

It is difficult to record early AMI data in patients. Currently, there is only one study in humans in which troponin change was examined after the LAD was occluded for up to 90 s [9]. Understanding the changes that occur in the early stage of AMI is useful in exploring the possibility of early AMI detection, as outcome depends on the time that elapses before treatment begins. The benefit of early treatment of MI is clear: Survival rates increase by up to 50% if reperfusion is achieved within one hour of symptom onset and by 23% if reperfusion occurs within 3 h [10,11,12]. Without immediate treatment, patients have higher rates of mortality and severe complications caused by increased infarct size, including cardiogenic shock, arrhythmias, and heart failure [13]. Currently, the two most commonly used methods to detect AMI—ECG and troponin changes—are available only in the hospital. Thus, the time it takes to diagnose an AMI depends on the patient’s awareness; the faster the patient contacts the hospital, then the earlier the AMI can be detected. Although educating patients about the symptoms of MI is highly recommended for increasing awareness, differentiating AMI from other diseases can be challenging, especially considering that up to one-third of patients may not have chest pain, which is a characteristic feature of MI [4]. Unfortunately, in the US, achieving an early AMI diagnosis in less than 3 h is difficult [6].

Wearable devices continue to be refined and offer a new way to monitor a wearer’s status by continuously and automatically measuring vital signs. When patients have an AMI, vital variables change immediately, as indicated in our preclinical model. A wearable device can detect these sudden changes, predict the risk of AMI based on an algorithm, and trigger an alert to notify the wearer. We believe this approach could dramatically reduce the time that elapses before individuals seek medical treatment at a hospital. Although this specific strategy has not been tested for AMI, several similar applications have been examined. For example, Biobeat, which has sensors that continuously monitor 13 physiological parameters including HR, blood pressure, cardiac output, and single-lead ECG, has been assessed in in-hospital patients for providing real-time monitoring, triggering alerts, and predicting patient deterioration [14]. Zio AT is an adhesive-patch EKG monitoring device that can continuously monitor ECG changes and alert patients when they have preset events based on the AI algorithm; this device has been shown to detect significantly more events than a conventional Holter device in ambulatory situations [15].

To achieve early AMI detection, a wearable device must first have the capacity to measure multiple key variables. Providing a more comprehensive view of the wearer’s situation not only increases the accuracy of the diagnosis but also decreases false positives. For the reliable detection of AMI and the exclusion of other types of myocardial injury such as tachycardia, our results suggest several pathophysiological changes that occur after AMI (Figure 5), such as blood pressure, ECG, and troponin/lactate levels, must be recorded. Although lactate is not currently used as an indicator for AMI detection, it has been correlated with severity and mortality in patients with AMI [16,17,18]. Thus, tracking lactate levels may offer valuable information in AMI prognosis and may help physicians adjust the treatment plan based on risk classification. Furthermore, data on the tracking of lactate levels in large populations offer the opportunity to explore further the relationship between lactate levels and AMI, which may provide additional value if combined with troponin changes. In our study, lactate levels increased within 15 min after LAD occlusion and presented a similar trend as the non-invasive troponin reading. If this lactate trend is verified in a large group of patients, the combination of lactate and troponin may further increase the accuracy of AMI detection. Table 1 summarizes the recording functions of popular wearable device brands on the market. Many brands listed are already equipped with the ability to record a single-lead ECG signal. Several products have been approved for blood pressure monitoring in other countries, but only two brands (Omeron and Biobeat) have received Food and Drug Administration (FDA) approval in the US. No commercial products currently available have the ability to detect troponin or lactate levels. Some prototypes are being tested, including the transdermal cardiac injury sensor we used in our study, which has shown good correlation with high-sensitivity cardiac troponin I blood samples in the clinic [19]. For lactate biosensors, Rockley’s Bioptx Pro platform has cooperated with Medtronic to start the application for clinical verification [20]. Other biosensors for blood troponin [21] and lactate [22] measurement are under development. We believe the hardware capacity is close to being able to achieve the target of detecting AMI in early stages.

A second consideration for devices is the need for all functions to run automatically. Without the need for wearer interference, compliance is increased. A comparison study of a regular Holter monitor with a novel ECG patch that patients did not need to manipulate suggested that the novel patch can achieve 98% patient compliance [23]. Higher compliance can improve the quality and accuracy of the recording and prevent patients from discontinuing the use of the device. Currently, almost all recording functions on wearable devices are automatic, except for ECG function.

For ECGs, most devices require that a wearer touch the device with a finger to start the recording [24], but some ECG patches do not need any intervention to record. However, regardless of the recording type, all offer only one-lead recording, which is insufficient for detecting AMI. The feasibility of Apple Watch’s ECG function in AMI detection has been studied [25,26,27,28,29]. Although it is possible to expand the ability of the Apple Watch to record an ECG from nine different leads (I, II, III, precordial V1 to V6), this approach is time consuming (5.7 min) [28] and requires education and practice to correctly record in nine positions [30]. These drawbacks could potentially decrease the wearer’s compliance. Some research has suggested using three rather than nine leads to record the ECG, but this strategy still requires education and active involvement of the wearer [31]. Thus, these limitations constitute a challenging hurdle to using ECG as an early detection tool on a wearable device. However, it may be possible to use blood pressure and biomarker levels to predict AMI risk and then have the wearer seek an ECG at the hospital, under the device’s instruction. This two-step prediction could decrease the required manipulations and increase the wearer’s compliance to improve intact data recording, thus enhancing accuracy. Sopic et al. created a unique, two-level classification system for MI detection. In this approach, an initial screening level uses only a few features to detect if any ischemic abnormalities need further evaluation; the second-level classifier is more computationally demanding but more accurate than the screening level [32]. In tests on SmartCardia INYU devices (SmartCardia, Switzerland), the algorithm achieved a clinically relevant accuracy of 90% for classifying MI [32].

Automatically and continuously recording multiple variables will help enable the wearable device to monitor all the data needed for the early detection of AMI. According to market research, an estimated 30% of US residents own a smart wearable device [33]. In the first quarter of 2022, Apple delivered more than 32 million Apple Watch devices to consumers [34]. The global market is expected to grow at a compound annual rate of 25%, reaching $70 billion by 2025 [35], and annual smartwatch shipments are projected to exceed 130 million units by 2025 [36]. Because of this expected large increase in the use of these devices, extensive data can be collected over time. Routine readings in individuals will serve as baseline data, and group data can be obtained from the large population of users. These data will dramatically increase the sample size for analyzing the diagnosis threshold for AMI. For example, the 2022 American Heart Association/American College of Cardiology guidelines for chest pain diagnosis indicated detecting changes in troponin levels in the very early stage of AMI is difficult [37]. This is due, in part, to the lack of sufficient sample size. In the TRAPID-AMI (The High Sensitivity Cardiac Troponin T Assay for Rapid Rule-out of Acute Myocardial Infarction) study, the mean time from symptom onset to presentation was 1.9 h. Still, it took 1.5 h after hospital presentation to obtain the first high-sensitive troponin sample [38]. The very limited data on early-stage troponin changes make it difficult to draw significant conclusions. However, wearable devices may be able to provide an extensive data pool for conducting meaningful analyses. In our study, the transdermal troponin sensor read the cardiac troponin I level every minute and provided many more readings during the balloon occlusion than other methods obtained through blood work. Our data generated an upward trend in the troponin curve with an obvious delta value: a clear sign of AMI. However, results from the blood tests (1 for every 15 min) did not show a delta during the occlusion. This finding directly indicates the benefits of large sample pools.

Large sample pools of both population and personalized data may be helpful in identifying more significant changes to develop an AI algorithm. This will help in generating auto-alerts for notifying patients about AMI risk. In a recent study, a machine learning algorithm was applied to analyze basic physiological, clinical, and paraclinical data from 150 patients. The algorithm obtained an MI prediction accuracy of 88.67% by using only a blood test, ECG readings, and echocardiography findings [39]. Although it is in early stages in AMI detection, the AI algorithm is booming in the field of arrhythmia diagnosis because of the large data pool. For example, the Fitbit heart study enrolled 455,669 participants [40], and the Apple heart study enrolled 419,927 participants [41]; both studies successfully verified the atrial fibrillation algorithm with a high positive predictive value based on this extensive dataset. The AI algorithm for predicting atrial fibrillation is now as accurate as the gold standard [42]. Zio, a wearable ECG device company, developed a deep-learning algorithm that could automatically identify 10 different arrhythmias accurately. This algorithm was developed based on 91,000 ECG recordings in 53,000 patients who used the Zio patch and has been approved by the FDA [43,44].

Although the future of wearable devices in AMI detection is promising, many challenges remain. Hardware capable of recording all the necessary functions is still under development. Furthermore, even after sufficient data are generated from the wearable devices, significant investment and resources are needed to design trials to collate, analyze, interpret, and finally develop efficient algorithms. Accurate algorithms will require extensive equipment and the enrollment of large numbers of patients. Moreover, after a practical algorithm is developed, another significant challenge is how to incorporate and transfer the wearable data into the current health system in compliance with existing policies and laws [45].

Our retrospective review has limitations. The major drawback was the small number of animals that underwent lactate and troponin testing, which limits the significance of the results. Our review included data only from pigs with AMI, which means we examined hemodynamic and biomarker changes only under conditions positive for AMI. For further study, we will include more preclinical models to compare how these variables change in different scenarios.

Despite the challenges, wearable devices can provide real-time monitoring of key indicators related to AMI. They could alert wearers early at the onset of an MI, which would change the current strategy of AMI diagnosis and dramatically reduce treatment delay.

## Figures and Tables

**Figure 1 diagnostics-13-01006-f001:**
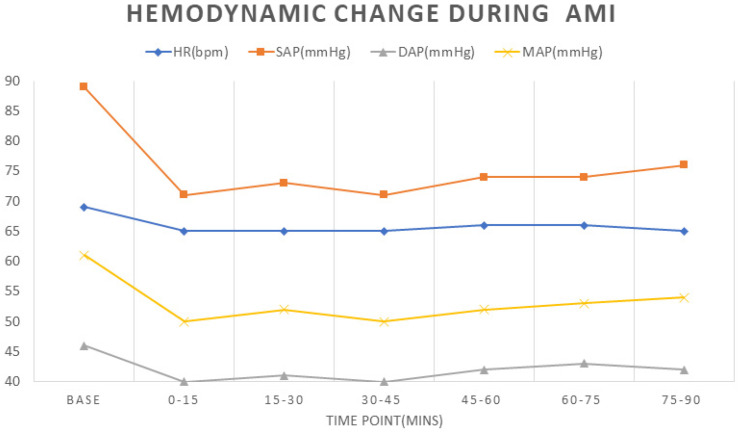
Heart rate and blood pressure at baseline and 15-min intervals during LAD occlusion in pigs. HR: heart rate; SAP: systolic arterial pressure; DAP: diastolic arterial pressure; LAD: left anterior descending artery; MAP: mean arterial pressure. Baseline was defined as the last reading before LAD occlusion.

**Figure 2 diagnostics-13-01006-f002:**
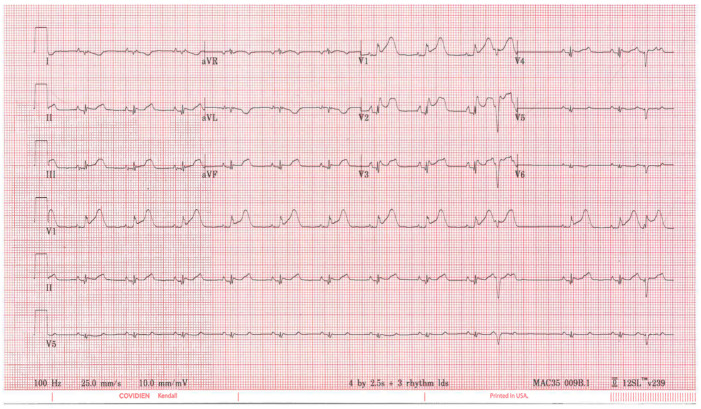
Representative 12-lead ECG recording at 5 min after LAD occlusion.

**Figure 3 diagnostics-13-01006-f003:**
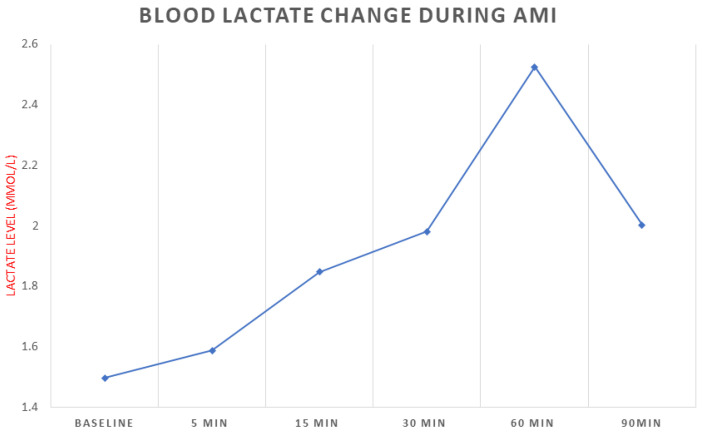
Blood lactate levels at baseline and during the occlusion period in pigs.

**Figure 4 diagnostics-13-01006-f004:**
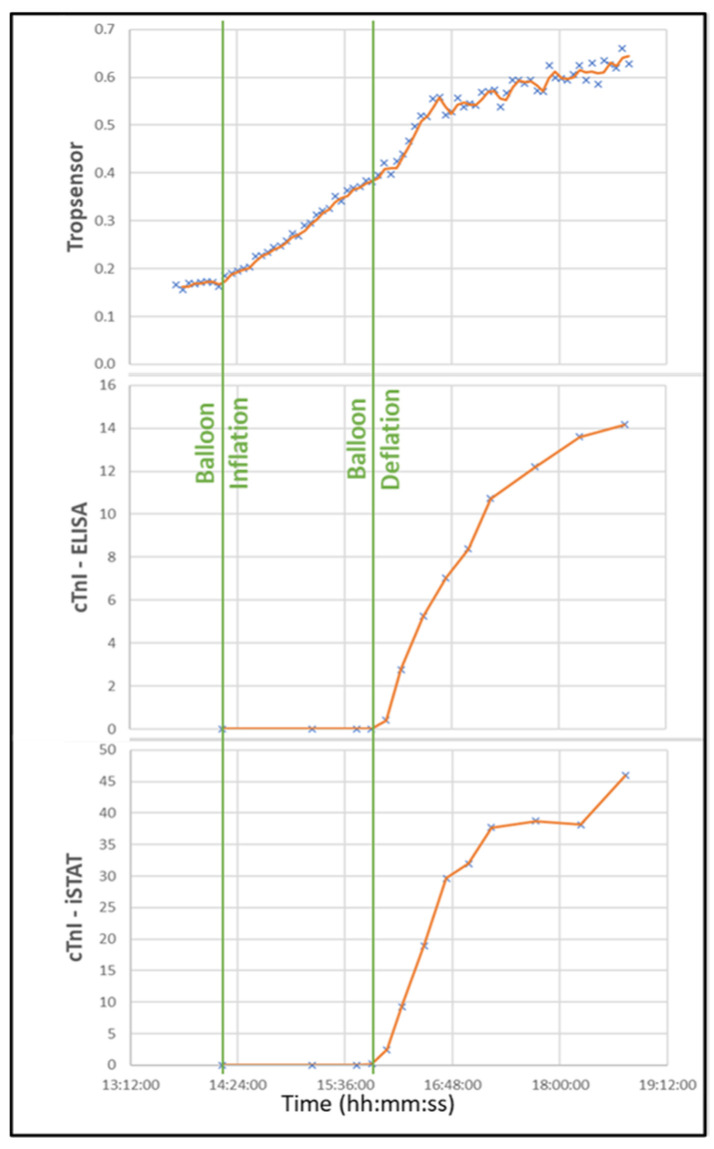
Real-time temporal correlation of the Tropsensor output with ELISA-based high sensitivity cardiac troponin 1 and IDEXX traditional cardiac troponin 1 measurements. The presence of infarction was confirmed by ST-segment elevation in EKGs performed at the set time points. When the LAD occlusion was resolved (balloon deflation), the influx of blood caused an increase in the troponin reading in all methods. cTn1, cardiac troponin 1.

**Figure 5 diagnostics-13-01006-f005:**
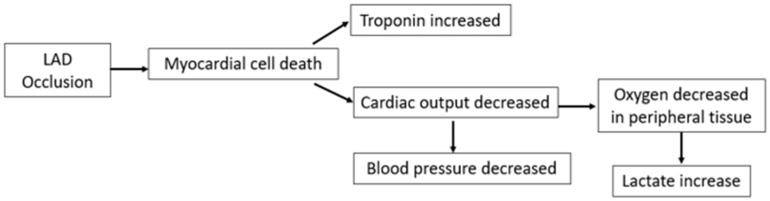
Acute changes after AMI onset. Schematic depicting acute hemodynamic and biomarker changes after an acute myocardial infarction was induced in pigs by occlusion of the left anterior descending artery (LAD).

**Table 1 diagnostics-13-01006-t001:** Summary of FDA-approved popular wearable devices on the market.

	Heart Rate	Blood Pressure	ECG
Apple Watch 8	√		√
Fitbit Sense 2	√		√
Google Pixel Watch	√		√
Samsung Galaxy Watch 5 *	√	√	√
Omeron+Alivcor Complete	√	√	√
Omron Heart Guide	√	√	
Biobeat BB-613WP	√	√	√ (with patch)
iRhythm Zio Patch	√		√
Zephyr BioHarness	√		√
Preventice Solutions Body Guardian	√		√
Corventist Nuvant MCT	√		√
BradyDx CAM	√		√
BioTel Heart	√		√
Medibio Sense MBS	√		√
Amazfit GTR 3 Pro *	√	√	

* Blood pressure function is not approved in the United States but is approved in other countries. Information verified as of 10/2022.

## Data Availability

The data are contained within the article.

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
