# Peer review of "Early Changes in Acute Myocardial Infarction in Pigs: Achieving Early Detection with Wearable Devices"

_diagnostics, 2023, doi:10.3390/diagnostics13061006_

Round 1

Reviewer 1 Report

This manuscript raises an important issue and provides relevant resultsIt is scientifically sound and appeals to a broad readership. But some remarks still remain.

1. It is common to refer to guidelines, especially in matters relating to the definition and diagnosis of myocardial infarction (MI). According to the current guidelines, the use of the troponin biomarker does not raise any questions. However, the use of lactate as a second biomarker seems doubtful. It is not recommended as a diagnostic marker for acute MI which is confirmed by the guide 2022 quoted by the authors. Data on the prognostic impact of hyperlactatemia mainly stem from observational investigations. It should be noted that observational studies do not allow to draw a conclusion on the causal relationship between lactate and clinical outcomes. This means that the use of a biomarker with insufficient clinical evidence is not quite appropriate.

2. However, as the work is pilot in nature, it is possible to use such a marker. But in this case, the authors should at least comment on this issue. They need to add a discussion about the importance of lactate for the diagnosis of acute MI and the feasibility of its use in the rapid diagnosis of a heart attack using a wearable device. Instead, they provided a unique reference [19] to a study on wearable lactate sensors for sweat analysis, which, in fact, does not fit the issue of acute MI.

3. In the “Results”, the standard deviation must be preceded by a “plus / minus” and not just a “plus”.

In conclusion, in spite of these minor comments, the article may be recommended for publication.

Author Response

Reviewer 1

This manuscript raises an important issue and provides relevant resultsIt is scientifically sound and appeals to a broad readership. But some remarks still remain.

  1. It is common to refer to guidelines, especially in matters relating to the definition and diagnosis of myocardial infarction (MI). According to the current guidelines, the use of the troponin biomarker does not raise any questions. However, the use of lactate as a second biomarker seems doubtful. It is not recommended as a diagnostic marker for acute MI which is confirmed by the guide 2022 quoted by the authors. Data on the prognostic impact of hyperlactatemia mainly stem from observational investigations. It should be noted that observational studies do not allow to draw a conclusion on the causal relationship between lactate and clinical outcomes. This means that the use of a biomarker with insufficient clinical evidence is not quite appropriate.

Response: We appreciate the Reviewer’s input and agree that lactate levels have not been used as a biomarker to detect AMI. We included the lactate data in our study for several reasons. First, lactate levels could indicate the severity of AMI. Second, we wanted to observe the trend of lactate to see if continuous lactate monitoring has any value in predicting AMI, as it has been a part of some wearable device platforms. Thus, we believe further examination of this topic may be warranted, and our study provided an opportunity to look into the value of lactate in this setting. As per the Reviewer’s comment, we have now added further explanation of this topic in the Discussion of the manuscript (lines 170-179). 

  1. However, as the work is pilot in nature, it is possible to use such a marker. But in this case, the authors should at least comment on this issue. They need to add a discussion about the importance of lactate for the diagnosis of acute MI and the feasibility of its use in the rapid diagnosis of a heart attack using a wearable device. Instead, they provided a unique reference [19] to a study on wearable lactate sensors for sweat analysis, which, in fact, does not fit the issue of acute MI.

Response: We apologize for the use of the inappropriate reference, and we have now updated the bibliography with several new supporting references (16-18) to reflect the importance of lactate in AMI. In addition, we have added a reference describing a new biosensor (22) that can track blood lactate level. As stated above, we have expanded our discussion on the importance of lactate in diagnosing AMI and on the feasibility of its use (lines 170-179).

  1. In the “Results”, the standard deviation must be preceded by a “plus / minus” and not just a “plus”.

In conclusion, in spite of these minor comments, the article may be recommended for publication.

Response: We apologize for the oversight in the signs used. We agree with the Reviewer that it is important to display the data properly, and we have now corrected the sign in the text.  

Reviewer 2 Report

This is a very interesting study about the early detection of acute myocardial infarction in pigs. The article is well structured and the results may have important clinical implications. The changes that occur during the ischemic cascade are well-known, and also the fact that detecting them as early as possible can prevent the occurrence of a fatal event. The authors presented the biological and electrocardiographic changes that can be diagnosed, but I would like to ask if they also evaluated the echocardiographic changes.  

Author Response

Reviewer 2

This is a very interesting study about the early detection of acute myocardial infarction in pigs. The article is well structured and the results may have important clinical implications. The changes that occur during the ischemic cascade are well-known, and also the fact that detecting them as early as possible can prevent the occurrence of a fatal event. The authors presented the biological and electrocardiographic changes that can be diagnosed, but I would like to ask if they also evaluated the echocardiographic changes.  

Response: We appreciate the Reviewer’s kind words and valuable comments. We evaluated the change in ejection fraction for most of the pigs but not during the 90 minutes of the AMI procedure. We performed a baseline echocardiogram before the AMI and did a follow-up echocardiogram on day 14 after the MI; we published those data in a separate paper cited below. We observed significant changes indicating left ventricular dysfunction and remodeling at 14 days after AMI. 

Li, K., Wagner, L., Moctezuma-Ramirez, A. et al. A Robust Percutaneous Myocardial Infarction Model in Pigs and Its Effect on Left Ventricular Function. J Cardiovasc Trans Res 14, 1075–1084 (2021).